# Effect of Wind Direction and Velocity on PV Panels Cooling with Perforated Heat Sinks

Sebastian Valeriu Hudișteanu *, Florin Emilian Țurcanu *, Nelu-Cristian Chereches *, Cătălin-George Popovici, Marina Verdeș, Diana-Ana Ancaș and Iuliana Hudișteanu

Department of Building Services, Faculty of Civil Engineering and Building Services, Gheorghe Asachi Technical University of Iasi, 700050 Iași, Romania

* Correspondence: sebastian.hudisteanu@tuiasi.ro (S.V.H.); emilian-florin.turcanu@tuiasi.ro (F.E.Ț.); chereches@tuiasi.ro (N.-C.C.)

**Abstract:** The numerical modeling of the effect of wind direction and velocity over the air cooling of PV panels with heat sinks is realized. During the study, a random PV panel with typical characteristics was analyzed for three different wind directions—towards its back, towards its front and from the side. The analysis was realized on a fixed PV panel, oriented to the south, with an inclination of 45 degrees from the horizontal position. The accuracy of the numerical simulation was achieved by comparison with the experimental studies presented in the literature and by comparing the NOCT conditions. The numerical study is focused on different types of heat sinks attached to a typical PV panel. The fins were distributed both horizontally and vertically. A challenging task consisted in simulation of the real wind conditions around the PV panel by taking into account the entire air domain. The simulations were realized for air velocity $v_{air}$ from 1 m/s to 5 m/s, solar radiation of $G = 1000$ W/m$^2$ and ambient temperature $t_{air} = 35$ °C. The output parameters analyzed were the average temperature of PV panels and their power production. Although the lowest temperatures were achieved for the back wind, the cooling effect was more intense for the side wind. The other direction studied also determined the cooling of PV panels. The passive cooling solutions analyzed introduced a rise of maximum power production between 1.85% and 7.71% above the base case, depending on the wind direction and velocity.

**Keywords:** CFD analysis; wind direction; PV temperature; air cooling; perforated heat sinks; NOCT validation

## 1. Introduction

The current global political, economic and energy context is characterized by high demand and consumption of population, instability, uncertainty and rampant inflation, right in the area of resources and energy [1]. Even without taking into account the current context, there is the well-known concept of Overshoot Day which is "celebrated" each year. In essence, this day represents the moment when humanity had already used up all of the renewable resources of the current year and unfortunately it takes place earlier and earlier each year. For the current year, this day was on 28 July 2022, meaning that from 29 July onwards, we started consuming additional resources compared to how much the planet could regenerate [2].

In this context, the use of solar-related energy could reduce the vulnerability of the European Community and eliminate the dependence of third-party oil or gas producers. Solar energy and its conversion into electricity by using photovoltaic (PV) panels became a major attraction in order to reduce the dependence on fossil fuels. After commissioning, there is obtained a clean and sustainable form of energy [2,3]. The photovoltaic systems could represent a stable and competitive area [4], taking into account both the recent actions in Europe [5] and the general tendency on the energy market that have quickly affected the stability of the world countries.

The advantages of photovoltaic technology are well documented in literature [6], but the control of the PV panels temperature is also an important challenge that could assure the enhancement of their efficiency [7].

The irreversible effect of heat generation in PV panels cannot be avoided, taking into account that the PV panel is constantly exposed to high levels of solar energy. Taking into account the efficiency of recently photovoltaic panels (of 20–25%), it can be stated that the vast majority of the remaining solar energy absorbed is converted into heat. This phenomenon, corroborated with the temperature-dependent variation of the efficiency of PV panels [7], determines a continuous concern on reducing the operating temperature. The data from literature present a decrease of the PV efficiency with the temperature between $-0.30$ and $-0.50\%/^{\circ}\text{C}$ [8,9]. It must be specified that, in contrast to STC (Standard Test Conditions), in normal operation the photovoltaic panels can reach temperatures of 70 °C or higher [8,10,11], at 1000 W/m$^2$, leading to a significant decrease in their efficiency.

It is well known that, when it is possible, it is recommended to use cooling solutions for improving PV panels' efficiency and develop PVT panels (photovoltaic–thermal) as a consequence [12–15]. There are considerable research studies in the literature regarding the cooling solutions for improving the efficiency of photovoltaic panels by reducing their operating temperature [6,9,10,13–15].

The air-cooling effect using heat sinks is tested for different areas: residential sector, urban and power plants [16,17].

The aim of this study was to analyze the effect of wind direction and velocity over different types of heat sinks attached to PV panels in comparison to the base case (without a cooling solution). The main target of the work was to quantify the cooling effect of various heat sinks in variable wind conditions. The presence of heat sinks is influencing the performance of PV panels, but the wind directions could also be an important factor in this regard. In contrast to the majority of similar studies in the literature, in this paper, the numerical modeling was conducted in more realistic wind conditions and inclination of the PV panel [18]. Taking into account the research from the literature, which concluded that the results for different temperature and radiation threshold are varying linearly for values of 20–35 °C and 600–1000 W/m$^2$, during this study, the most unfavorable conditions in terms of PV panel heating were detailed: solar radiation $G$ = 1000 W/m$^2$ and ambient temperature $t_{air}$ = 35 °C [6].

## 2. Related Research

There are numerous related research studies in the literature, both experimental and numerical, regarding the inverse dependence of the temperature of monocrystalline and polycrystalline PV panels with their efficiency, and consequently the necessity of cooling solutions for removing this shortcoming [6,8,9,11,19,20].

Related research works on PV panels' cooling by using air are presented in the literature, and a large number of technologies and solutions to improve their efficiency are presented [9,21–26]. Recent experimental and numerical studies regarding the passive air cooling of PV panels by using heat sinks [8,27,28] tend to confirm noticeable and promising results [6,8,22,23]. The two main materials used for manufacturing the heat sinks are aluminum and copper [23]. The cooling effect is analyzed through experimental tests under different conditions and the conclusion is that copper has superior results [23]. Also, numerical and experimental analysis showed that the presence of perforations on the fins of the heat sinks can improve the efficiency of the PV panels by reducing their operating temperature [6,23]. The predicted temperature drop, due to the use of these solutions, can reach values between 7 °C and 10 °C [23].

Numerical simulations regarding the wind direction towards the back, front and side of heat sinks attached to PV panels showed a potential of operating temperature reduction towards 10 °C below the basic case temperature (when no cooling is applied) [8,11,15,29,30]. The improvement of the PV panels' performance was determining a rise of normalized power production up to 0.90 of the standard test conditions one [31]. As an average, the

normalized power generation of photovoltaic panels can reach about 0.70–0.83 compared to the STC rating [18].

## 3. Materials and Methods

### 3.1. Problem Description

The study was focused on determining the effect of wind direction and velocity over the resulting operating temperatures of PV panels both in cooling and non-cooling conditions (base case). Taking into account the variety of PV panels on the market, a typical one is analyzed (with a power production of 320 $W_p$ and a surface of 1.6 $m^2$). The PV panel is a monocrystalline Si-based one—AE320HM6-60—Table 1 [32], inclined at 45° from the horizontal position, in constant solar radiation $G$ = 1000 $W/m^2$, constant ambient temperature of $t_{air}$ = 35 °C and variable wind direction (from the back, front and side of the PV panel) and velocity (from 1 m/s to 5 m/s). This configuration is also used for validation of the model in NOCT (Nominal Operating Cell Temperature) conditions [31].

**Table 1.** The electrical and dimensional characteristics of the photovoltaic panel studied.

| AE Solar AE320HM6-60 PV Panel [32] | |
| --- | --- |
| Maximum power $P_{mp}$ | 320 $W_p$ |
| Voltage at maximum power $V_{mp}$ | 33.40 V |
| Current at maximum power $I_{mp}$ | 9.59 A |
| Open-circuit voltage $V_{oc}$ | 40.90 V |
| Short-circuit current $I_{sc}$ | 10.15 A |
| Efficiency $\eta_{STC}$ | 19.30% |
| Temperature factor of efficiency $\beta$ | −0.37%/°C |
| Nominal operating cell temperature $t_{NOCT}$ | 47 °C |
| Dimensions of PV panel (length $L$ × width $l$ × height $h$) | 1665 mm × 996 mm × 35 mm |

During the study, there were different stages of analysis of the heat sinks attached to the PV panel. The study was initiated with the validation of the model for the base case (no cooling system attached to the photovoltaic panel). After that, a total of six configurations of heat sinks were attached to the PV panel and studied. The various heat sinks were obtained by changing the configuration of their fins. Therefore, for each type of fins (horizontal and vertical), there were the following cases: non-perforated fins vs. fins with perforations of 30 mm and 60 mm diameter ($\Phi$).

The proposed heat sinks are placed in the backside of the PV panel. The contact region is represented by a copper sheet (plate). Perforated or non-perforated copper fins are connected to this plate in order to improve the heat transfer by amplifying the turbulence from the PV panel to the surrounding air (see Figure 1). Therefore, the perforations assure a quantitative and qualitative effect against the heating of the PV panel and could improve the photovoltaic efficiency.

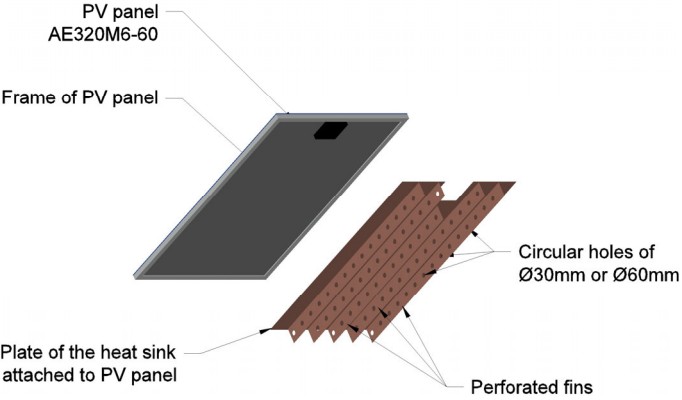

**Figure 1.** The PV panel—heat sink ensemble—Case of vertical fins with perforations.

The literature analysis showed that the circular holes made in the fins of the heat sinks have the potential to produce promising results for cooling the PV panels [6,8,22,23,29]. In this case, the resulting temperatures are lower by 2–3 °C than those obtained by using other shapes for perforations [18]. In the literature are presented various types of fins (continuous, discontinuous or staggered) [33]. During this study, continuous fins of 100 mm height were analyzed.

Two main configurations of the heat sinks studied were:

Model 1—**horizontal** fins (3 models: non-perforated, perforated with $\Phi = 30$ mm holes and $\Phi = 60$ mm holes);

Model 2—**vertical** fins (3 models: non-perforated, perforated with $\Phi = 30$ mm holes and $\Phi = 60$ mm holes).

*3.2. Numerical Modeling*

The ANSYS Fluent platform was used for numerical modeling in steady-state conditions. The geometric model and the mesh were created by using ANSYS Fluent-Meshing and ANSYS SpaceClaim.

Given the particularities of the flow (that is not aligned with the mesh), the simulations were realized in the SIMPLE method. The second-order accuracy was used for solving the governing equations of the phenomena involved [8,34].

The solar radiation was modeled by the aid of the native module of Fluent—Solar Ray Tracing. The conversion of solar radiation into heat is synthesized as an absorption coefficient of the PV panel of $\alpha = 0.7$ [6,8].

The energy balance between the PV panel–heat sink ensemble and the surroundings were modeled by including both the convection and radiation effects [28]. The turbulence model used for the flowing air was the $k$–$\varepsilon$ realizable model [34].

The geometry of the model was created using SpaceClaim. For all studied cases, after preliminary analysis, the distance between fins was imposed to $s = 150$ mm. The fins are orthogonal to the heat sink sheet, as shown in Figure 1. The circular perforations had 30 mm or 60 mm in diameter and they are located at 100 mm one to another according to previous research works [6,23]. In this way, a better air circulation is achieved near the heat sink and the cooling effect is enhanced.

It is well known that the accuracy and correctness of the mesh lead to precise CFD (Computational Fluid Dynamics) [34]; therefore, in order to solve the highly orthogonal mesh that could result from a conventional mesh. The polyhedral mesh for surfaces and poly-hex mesh for volumes determined an increase of the simulation time, maintaining the same precision of the results. In order to achieve an appropriate mesh, different refinements are necessary [34]. For the present numerical modeling, the PV panel, heat sink and air volume had a mesh quality as presented in Table 2.

**Table 2.** The mesh quality of the simulation domain.

| Mesh Quality | |
|---|---|
| Aspect ratio | 5.0–60 |
| Skewness | 0.4–0.7 |
| Orthogonal quality | 0.1–0.3 |

In order to optimize the number of the cells of the mesh and the simulation time, a mesh independence study was realized. The preliminary simulations showed that $3 \times 10^6$ cells for the base case and $5.5 \times 10^6$ cells for cases with heat sinks determined accurate results compared to a higher number of elements (see Figure 2).

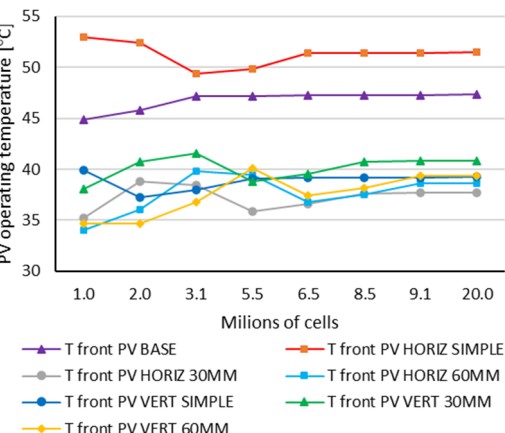

**Figure 2.** Mesh independence study for the numerical simulation.

*3.3. Sensitivity Analysis of Grid Spacing*

The spatial discretization error analysis was also performed. The Grid Convergence Index (*GCI*) method was used, according to [35,36]. The *GCI* represents a very accurate method used to evaluate how far the numerical computed values are compared to the asymptotic one [36]. The analysis was implemented on both base and heat sink cases. The following equations present the calculations used and the detailed values obtained. Three grid refinement levels were considered and the following *GCI* values were calculated:

$$GCI_{1-2} = \frac{F_s|\varepsilon_{1-2}|}{\left(r_{1-2}^p - 1\right)} \tag{1}$$

and

$$GCI_{2-3} = \frac{F_s|\varepsilon_{2-3}|}{\left(r_{2-3}^p - 1\right)} \tag{2}$$

where:

$GCI_{1-2}$—Grid Convergence Index for fine and medium grids;
$GCI_{2-3}$—Grid Convergence Index for coarse and medium grids;
$F_s$—Factor of safety ($F_s$ = 1.25 in the present study);
$\varepsilon$—Relative error:

$$\varepsilon_{n-n+1} = \frac{f_{n+1} - f_n}{f_n} \tag{3}$$

$p$—Order of convergence:

$$p = \frac{ln\frac{f_{n+2}-f_{n+1}}{f_{n+1}-f_n}}{\ln\ r} \tag{4}$$

$f$—Significant parameter chosen for the comparison ($f_1$—the temperature calculated with the finest grid, $f_2$—with the medium grid, $f_3$—with coarsest one);
$r$—Constant refinement ratio ($r$ = 2 in the present study):

$$r_{n-n+1} = \frac{h_{n+1}}{h_n} \tag{5}$$

$h$—Grid spacing, with:

$$h_1(= 1\ mm) < h_2(= 2\ mm) < h_3(= 3\ mm) \tag{6}$$

For the present study, the $f$ parameter consists in the average operating temperature of the PV panel, $t_{PV}$.

In order to achieve precise results from numerical simulations by using the lowest number of cells, the following auditing ratio should be as close as possible to 1:

$$\frac{GCI_{2-3}}{r^p GCI_{1-2}} \to 1 \tag{7}$$

In the case of using the finest grid, the predicted value at $h = 0$ is:

$$f = f_1 + \frac{f_1 - f_2}{r_{1-2}^p - 1} \tag{8}$$

For the specific values of the base case analysis ($r = 2, f_1 = 47.28, f_2 = 47.40, f_3 = 47.38$, $F_s = 1.25$), the following values were obtained: the order of convergence $p = 1.68$ and the predicted value $f = 47.31$. Also, $\varepsilon_{1-2} = 0.053\%$ and $\varepsilon_{2-3} = 0.169\%$, while the $GCI_{1-2} = 0.030\%$ and $GCI_{2-3} = 0.096\%$. The calculated auditing ratio is:

$$\frac{GCI_{2-3}}{r^p GCI_{1-2}} = 0.9977 \tag{9}$$

For the specific values of the heat sink case analysis ($r = 2, f_1 = 39.39, f_2 = 39.44$, $f_3 = 39.49, F_s = 1.25$), the following values were obtained: The order of convergence $p = 0.17$ and the predicted value $f = 39.80$. Also, $\varepsilon_{1-2} = 0.116\%$ and $\varepsilon_{2-3} = 0.130\%$, while the $GCI_{1-2} = 1.139\%$ and $GCI_{2-3} = 1.281\%$. The calculated auditing ratio is:

$$\frac{GCI_{2-3}}{r^p GCI_{1-2}} = 0.9975 \tag{10}$$

Therefore, the main verification of the grid quality confirms that the results are in the asymptotic range of convergence and the numerical model was further conducted by using the lowest cell number (coarse grid).

A large number of research studies from the literature are analyzing the photovoltaic panels either in the horizontal [19] or vertical [8,23] position. In addition, the surrounding air is most often simulated as a channel behind the photovoltaic panel [18]. Moreover, studies on cooling inclined photovoltaic panels are present in the literature, but in reduced number [6,11,27].

During this study, the numerical simulations were done considering the air circulation in the vicinity of an inclined PV panel cooled with heat sinks. It was possible due to the high accuracy mesh, which assured a realistic simulation of the airflow and heat transfer from the PV panel to the heat sink and to the surrounding air. In the present study, in order to simplify the numerical model, the influence of the relative humidity of the air was neglected. However, according to literature, the variation of the relative humidity could affect the final results by 4–5% [37]. The thermophysical properties of the materials defined for the simulations are presented in Table 3. The geometry and set-up of the simulation are shown in Figure 3.

**Table 3.** Thermophysical properties of the materials defined.

| Material | $g$ [mm] | $\lambda$ [W/m·K] | $\rho$ [kg/m$^3$] | $c_p$ [J/kg·K] |
|---|---|---|---|---|
| Glass | 3.00 | 1.00 | 2300 | 0.50 |
| PV cells | 0.35 | 168 | 2330 | 0.757 |
| EVA | 0.50 | 0.35 | 960 | 2.09 |
| Tedlar | 0.20 | 0.20 | 1500 | 1.20 |
| Copper | 1.00 | 386 | 8960 | 0.376 |
| Air | - | 0.025 | 1.20 | 1.005 |

where:

$g$—thickness of the layers composing the PV panel [m];

$\lambda$—thermal conductivity of the layers composing the PV panel [W/m·K];
$\rho$—density of the layers composing the PV panel [kg/m$^3$];
$c_p$—specific heat of the layers composing the PV panel [J/kg·K].

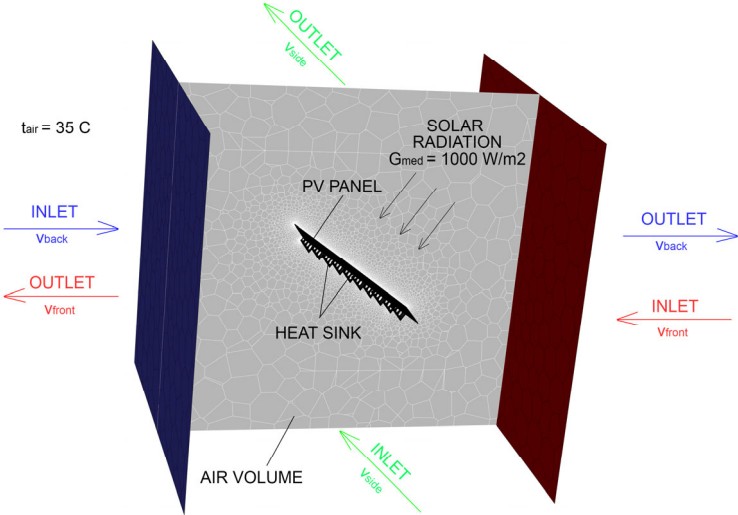

**Figure 3.** The geometry and set-up of the numerical simulation domain.

The steady-state conditions were assumed for the numerical modeling. During the simulations, the following inputs were established:

- **constant**: ambient temperature, $t_{air}$ = 35 °C—worst scenario in terms of heating [6];
- **constant**: solar radiation, $G_{med}$ = 1000 W/m$^2$—worst scenario in terms of heating [6];
- **variable**: velocity of the air, $v_{air}$ = 1–5 m/s;
- **variable**: wind direction: towards the back, front and side of the PV panel;

The main results obtained from the simulations were:

- operating temperature of the photovoltaic panel;
- normalized power production.

In terms of wind direction, the simulations were made in 3 Scenarios—for each case, different faces of the volume represent the inlet and outlet of the flow—Figure 3. The blue arrows are used for the back wind. The red arrows are used for the front wind. The green arrows are used for the side wind.

The inlet face of the domain varies as the position but has the same boundary conditions for each case: constant air temperature of 35 °C, velocity vector normal to the inlet surface and variable wind velocity: from 1 m/s to 5 m/s. The boundary conditions regarding the flow were the turbulent intensity of 5% and turbulent viscosity ratio of 10.

### 3.4. Studied Cases

A total of seven cases were studied. The first one is the base case (Case 0), while the other six consist of the heat sinks analyzed:

| | |
|---|---|
| 0. Base case | - no cooling; |
| 1. Horiz Simple | - heat sink and non-perforated horizontal fins; |
| 2. Horiz 30 mm | - heat sink and perforated $\Phi$ = 30 mm horizontal fins; |
| 3. Horiz 60 mm | - heat sink and perforated $\Phi$ = 60 mm horizontal fins; |
| 4. Vert Simple | - heat sink and non-perforated vertical fins; |
| 5. Vert 30 mm | - heat sink and perforated $\Phi$ = 30 mm vertical fins; |
| 6. Vert 60 mm | - heat sink and perforated $\Phi$ = 60 mm vertical fins. |

The analysis was realized on a fixed PV panel, oriented to the south, with an inclination of 45° from the horizontal position, and wind direction was varied towards the back, front and side of it, as shown in Figure 4.

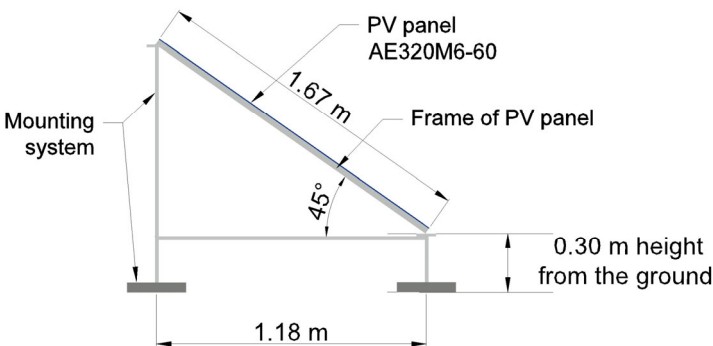

**Figure 4.** Side view of the set-up of the fixed PV panel, inclined at 45°.

The designed heat sinks consisted of a rectangular metal (copper) sheet with horizontal or vertical equidistant fins (Table 4). The main geometric characteristics of the heat sinks are: $h_{fins}$ = 100 mm (height of all fins implemented), $L_{fins}$ = 1520 mm (length of vertical fins) and $L_{fins}$ = 860 mm (length of vertical fins)—Table 4.

**Table 4.** The perspective view of the heat sinks attached to the PV panel.

| Case | 3D Perspective View |
|------|---------------------|
| 1–3 | 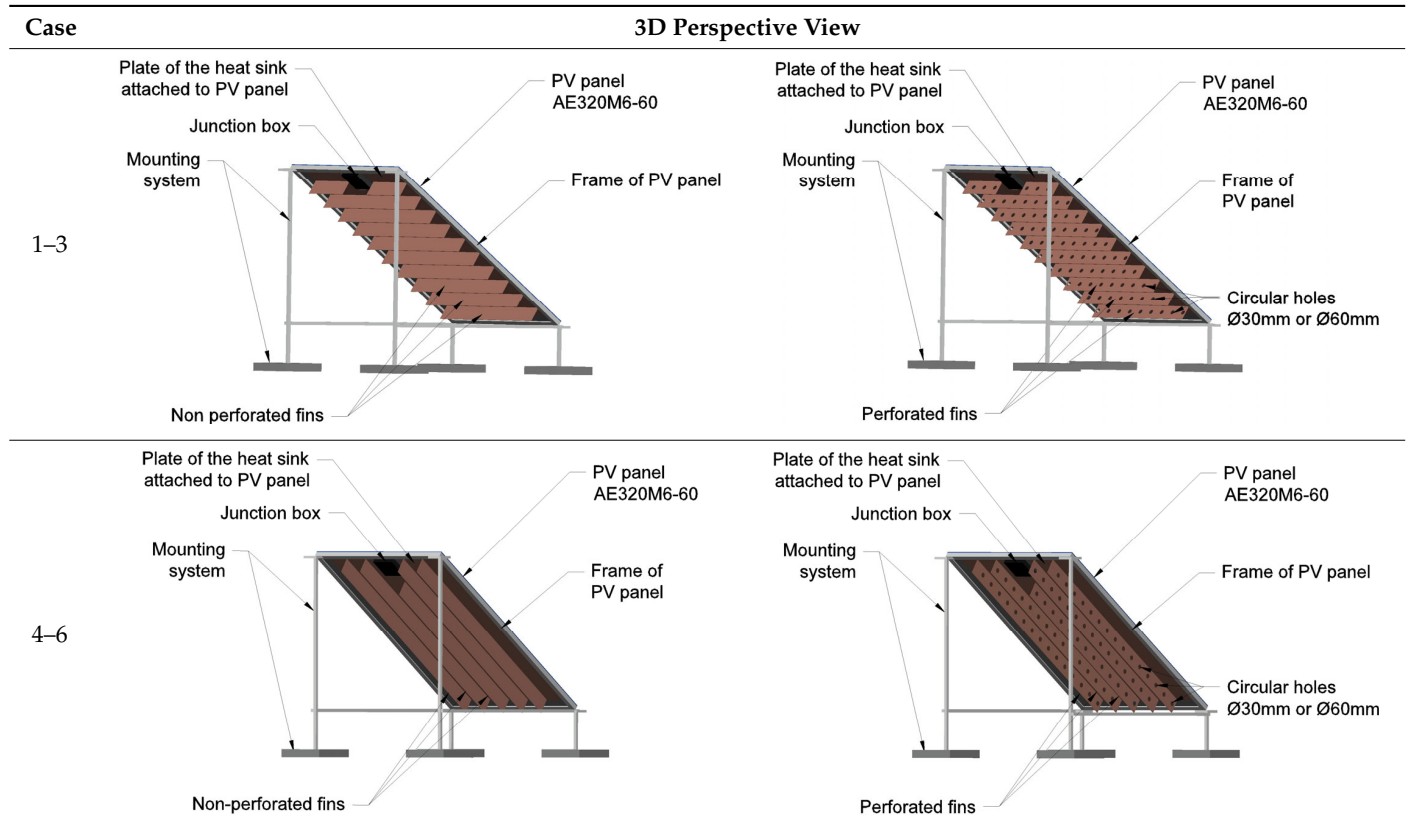 |
| 4–6 | |

The fins, made of copper, are studied for both perforated and non-perforated cases. The base sheet of the heat sink is in direct contact with the backside of the PV panel. The diameter of the circular perforations of the fins were of 30 mm or 60 mm (Table 4). The holes were placed at an equal distance of $d$ = 100 mm from one to another. In order to assure good air circulation, there was imposed an offset of 50 mm between two successive fins.

The main cases of cooling solutions analyzed are presented in Table 4. Cases 1–3 represent the horizontal (transversal) ones with no perforations or with perforations of 30

mm and 60 mm. Cases 4–6 are the vertical (longitudinal) ones, both with and without perforations.

## 4. Numerical Model Validation

### 4.1. Base Case Validation—Without Heat Sink

The base case was validated in two steps. The first one consisted in analyzing the behavior of the numerical model in NOCT conditions compared to the expected results according to the producer datasheet and literature. The NOCT conditions suppose a PV panel inclined at 45°, direct solar radiation on front of the PV panel of 800 W/m$^2$, air temperature of 20 °C and the wind velocity of 1 m/s towards the backside of the PV panel [31]. According to the producer of the PV panel [32], in these conditions, the average temperature of the PV panel $t_{NOCT}$ should be 47 °C. Taking into account that the good matching of the PV temperature from simulation was 47.16 °C, the model was considered appropriate and the next steps of validation were covered.

A common Equation (1) from literature [31,38] extrapolates the NOCT results and makes the connection between the PV panel temperature ($t_{PV}$) and outside conditions (air temperature $t_{air}$ and solar radiation G) for each PV panel, characterized by its own $t_{NOCT}$. By comparing the numerical results with those obtained with Equation (11), the following tendency of the PV panel temperature was observed (Figure 5).

$$t_{PV} = t_{air} + \frac{t_{NOCT} - 20}{800} \cdot G \qquad (11)$$

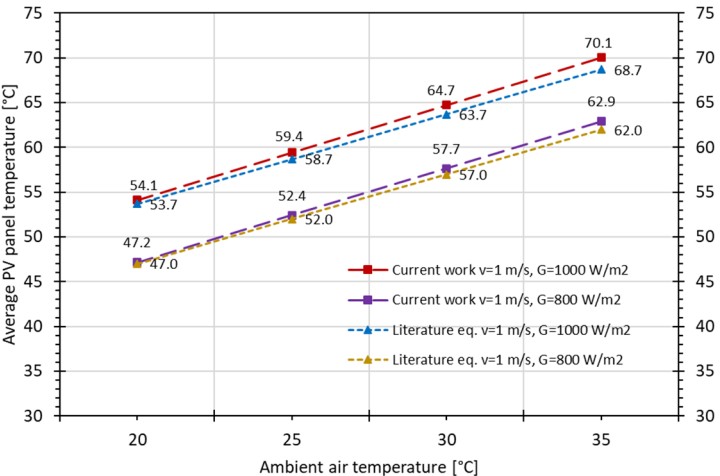

**Figure 5.** Validation of the numerical model using NOCT conditions and Equation (1).

The differences between the numerical model and Equation (1) were between 0.28 and 1.98%, which represents a very good correlation between them.

Also, experimental studies from literature [11,39] were compared with the present numerical simulation on the base case for different input variables. For the PV panel inclined to 45°, wind velocity of $v$ = 1.5 m/s and solar radiation of $G$ = 837 W/m$^2$, the difference between the present study (53.68 °C) and the experimental data (51.6 °C) [11] was 3.87%.

Additional results of simulation and experimental studies [23,39] are compared in Figure 6, and a good correlation can be found.

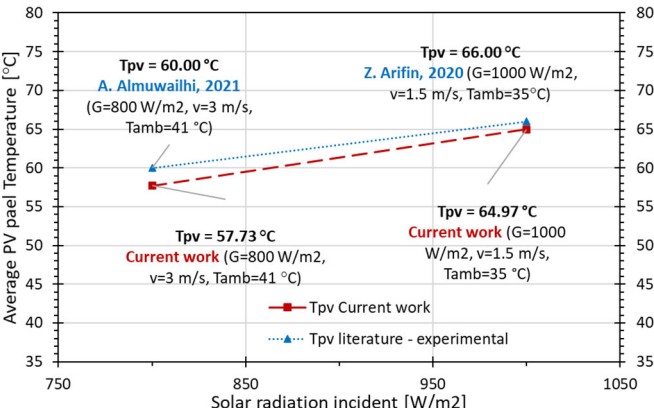

**Figure 6.** Validation of the numerical simulation—base case—PV panel without heat sink.

*4.2. PV Panel and Heat Sink Validation*

The heat sinks attached to a PV panel were also analyzed for validation. The comparison between experimental data [34] and the present model is presented in Figure 7.

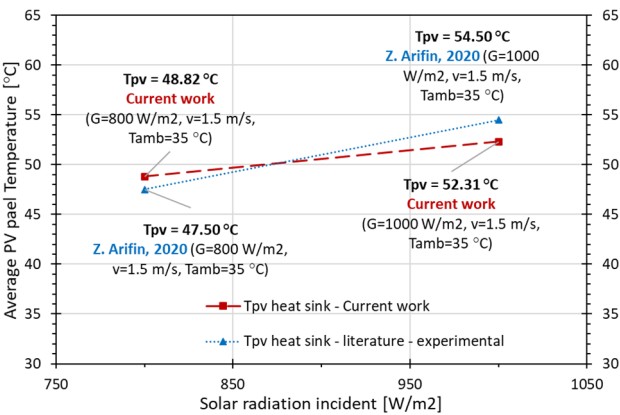

**Figure 7.** Validation of the numerical simulation—PV panel with heat sink.

In order to improve the accuracy, the model validation was realized also for various values of solar radiation and air temperature. The differences between the present numerical model and experiments from literature were from −5.18% to +4.03%. The narrow range of variation is considered acceptable [30], supporting the validity of the study. Therefore, the numerical modelling was considered to have little risk of error and the study was continued.

## 5. Results and Discussion

The present simulations have a higher level of accuracy compared to the majority of literature, because the air circulation around the PV panel–heat sink ensemble causes a change in temperatures distribution and non-homogeneous heat transfer. While the majority of existent studies are realized by using a constant convective heat transfer coefficient ($h_c$) in order to replace the air circulation around an object [18], in the present paper, the airflow near the photovoltaic panel and heat sink ensemble is also considered. This approach produces more realistic results and could represent a good starting point for future research in the literature, representing the connection between air velocity, direction and the power production obtained for these conditions.

The PV panel operating temperature and normalized power production were obtained. In Figures 8–10, there are presented qualitative and quantitative results regarding the average temperatures of the photovoltaic panels obtained for each model of the heat sink and various wind velocities and directions. These results are obtained by varying the set-up: $t_{air}$ = 35 °C, $v_{air}$ = 1–5 m/s and G = 1000 W/m$^2$ and three wind directions (Table 5).

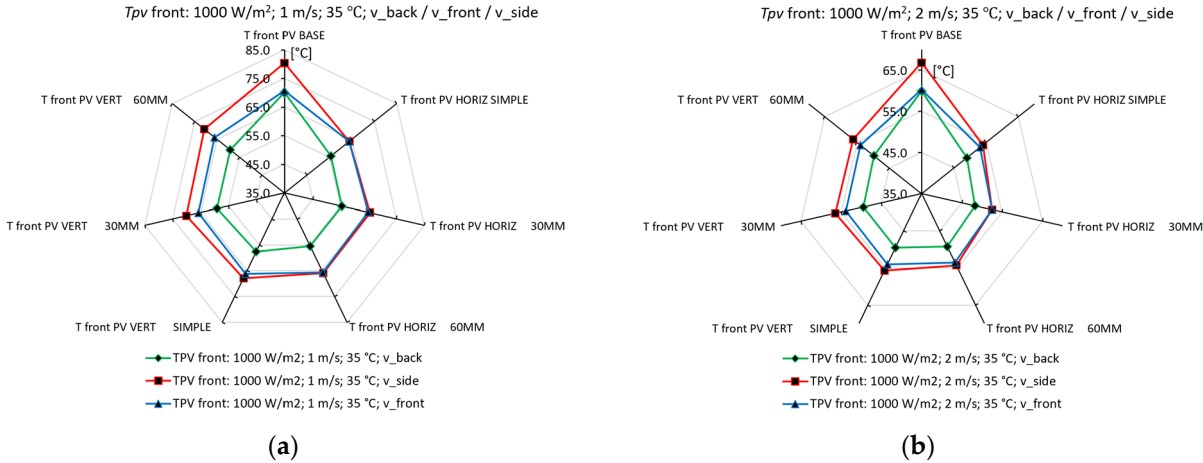

**Figure 8.** Variation of average operating temperature for (**a**) $v_{air} = 1$ m/s; (**b**) $v_{air} = 2$ m/s.

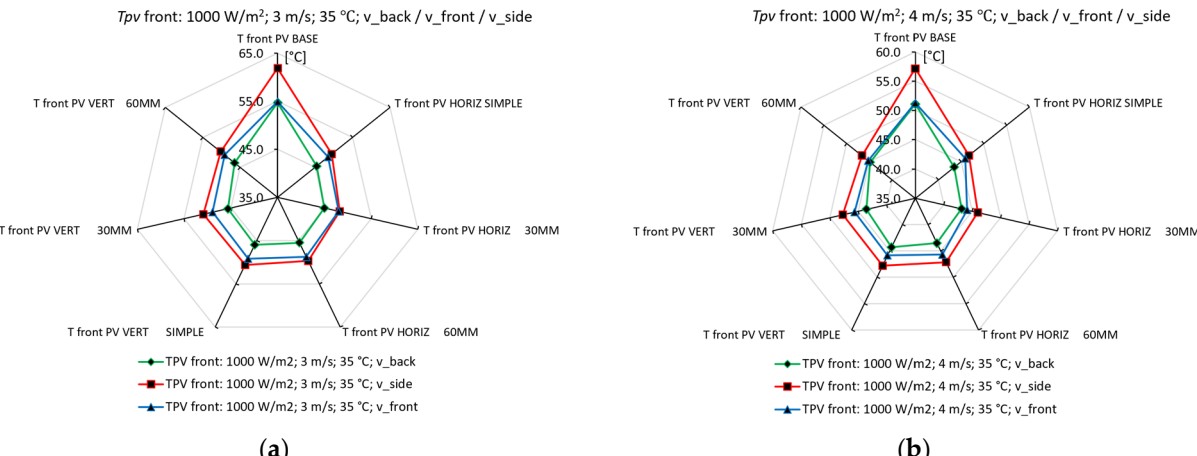

**Figure 9.** Variation of average operating temperature for (**a**) $v_{air} = 3$ m/s; (**b**) $v_{air} = 4$ m/s.

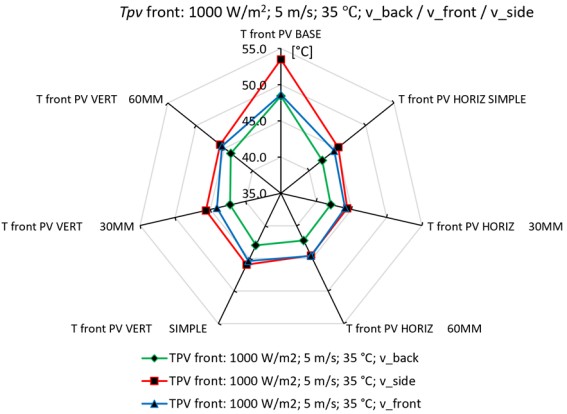

**Figure 10.** Variation of average operating temperature for $v_{air} = 5$ m/s.

**Table 5.** Operating temperature of PV panel for studied cases.

| Case | $v_{air}$ | Back Wind | | Front Wind | | Side Wind | |
|---|---|---|---|---|---|---|---|
| | | Drop of $t_{PV}$ Compared to Base Case, for $t_{air}$ = 35 °C | | | | | |
| | | °C | % | °C | % | °C | % |
| 0. Base Case | | 70.1 °C | - | 70.7 °C | - | 80.4 °C | - |
| 1. Horiz Simple | | −14.46 °C | −20.6% | −6.88 °C | −9.7% | −16.56 °C | −20.6% |
| 2. Horiz 30 mm | 1 m/s | −14.61 °C | −20.9% | −5.85 °C | −8.3% | −14.89 °C | −18.5% |
| 3. Horiz 60 mm | | −14.54 °C | −20.8% | −5.03 °C | −7.1% | −14.46 °C | −18% |
| 4. Vert Simple | | −12.4 °C | −17.7% | −4.42 °C | −6.2% | −12.54 °C | −15.6% |
| 5. Vert 30 mm | | −10.93 °C | −15.6% | −4.85 °C | −6.9% | −10.27 °C | −12.8% |
| 6. Vert 60 mm | | −10.87 °C | −15.5% | −4.53 °C | −6.4% | −9.73 °C | −12.1% |
| 0. Base Case | | 59.9 °C | - | 60.2 °C | - | 66.8 °C | - |
| 1. Horiz Simple | | −10.88 °C | −18.2% | −7.14 °C | −11.9% | −12.86 °C | −19.3% |
| 2. Horiz 30 mm | 2 m/s | −11.7 °C | −19.5% | −7.71 °C | −12.8% | −14.28 °C | −21.4% |
| 3. Horiz 60 mm | | −10.68 °C | −17.8% | −6.65 °C | −11% | −12.53 °C | −18.8% |
| 4. Vert Simple | | −10.38 °C | −17.3% | −6.17 °C | −10.2% | −11.15 °C | −16.7% |
| 5. Vert 30 mm | | −10.42 °C | −17.4% | −6.42 °C | −10.7% | −10.33 °C | −15.5% |
| 6. Vert 60 mm | | −10.13 °C | −16.9% | −6.27 °C | −10.4% | −10.67 °C | −16% |
| 0. Base Case | | 54.7 °C | - | 54.9 °C | - | 61.9 °C | - |
| 1. Horiz Simple | | −9.33 °C | −17.1% | −6.45 °C | −11.8% | −12.46 °C | −20.1% |
| 2. Horiz 30 mm | 3 m/s | −9.7 °C | −17.7% | −6.85 °C | −12.5% | −13.59 °C | −22% |
| 3. Horiz 60 mm | | −9.2 °C | −16.8% | −6.13 °C | −11.2% | −12.14 °C | −19.6% |
| 4. Vert Simple | | −8.69 °C | −15.9% | −5.64 °C | −10.3% | −11.25 °C | −18.2% |
| 5. Vert 30 mm | | −9.04 °C | −16.5% | −5.94 °C | −10.8% | −10.99 °C | −17.8% |
| 6. Vert 60 mm | | −8.21 °C | −15% | −5.75 °C | −10.5% | −11.64 °C | −18.8% |
| 0. Base Case | | 51.1 °C | - | 51.3 °C | - | 57.2 °C | - |
| 1. Horiz Simple | | −7.57 °C | −14.8% | −5.36 °C | −10.5% | −10.43 °C | −18.2% |
| 2. Horiz 30 mm | 4 m/s | −7.94 °C | −15.5% | −7.21 °C | −14.1% | −11.21 °C | −19.6% |
| 3. Horiz 60 mm | | −7.58 °C | −14.8% | −5.59 °C | −10.9% | −10.03 °C | −17.5% |
| 4. Vert Simple | | −6.8 °C | −13.3% | −5.41 °C | −10.6% | −9.4 °C | −16.4% |
| 5. Vert 30 mm | | −7.56 °C | −14.8% | −5.62 °C | −11% | −9.49 °C | −16.6% |
| 6. Vert 60 mm | | −6.24 °C | −12.2% | −5.98 °C | −11.7% | −10.52 °C | −18.4% |
| 0. Base Case | | 48.5 °C | - | 48.6 °C | - | 53.5 °C | - |
| 1. Horiz Simple | | −6.15 °C | −12.7% | −4.23 °C | −8.7% | −8.36 °C | −15.6% |
| 2. Horiz 30 mm | 5 m/s | −6.41 °C | −13.2% | −4.57 °C | −9.4% | −9.12 °C | −17% |
| 3. Horiz 60 mm | | −6.21 °C | −12.8% | −4.03 °C | −8.3% | −9 °C | −16.8% |
| 4. Vert Simple | | −5.47 °C | −11.3% | −3.21 °C | −6.6% | −7.57 °C | −14.1% |
| 5. Vert 30 mm | | −6.26 °C | −12.9% | −4.57 °C | −9.4% | −7.87 °C | −14.7% |
| 6. Vert 60 mm | | −4.61 °C | −9.5% | −3.21 °C | −6.6% | −7.78 °C | −14.5% |

These values could represent a good marker in order to choose, when possible, the right positioning of the PV panels related to the dominant wind in a certain geographical area. The best temperatures of the heat sinks were obtained when the wind was towards the heat sink—in the backside of the PV panel—determining a decrease of the average

operating temperature of approximately 15 °C. On the other hand, the best cooling effect was registered for the side wind, due to the higher temperatures resulted in the base case—when no cooling is applied. Although the lowest temperatures were achieved for the back wind, the cooling effect was more intense for the side wind. If, in the case of wind velocity, the effect of cooling evolves almost linearly from 1 m/s to 5 m/s, in terms of the wind direction, there is a slightly different behavior. Therefore, there are present some recirculation areas that modify the distribution of velocities and temperatures on the PV panel–heat sink ensemble. The worst wind direction, from this point of view, is the side one, causing a non-uniform temperature distribution, which is influencing the entire PV panel efficiency.

The quantitative results for $v_{air}$ = 1 m/s, air temperature $t_{air}$ = 35 °C and solar radiation $G$ = 1000 W/m$^2$ for each case are presented in the following graphs: Figures 8–10. In these figures, the values displayed on the plots (35, 45, 55, 65, 75 and 85) represent the scale for the average operating temperatures of the PV panel measured in °C. The tendency registered for 1 m/s is also manifesting for higher velocities: 2–5 m/s, with the highest temperatures recorded for the base case (T front PV BASE) and the lowest ones for the horizontal heat sink with 30 mm perforations (T front PV HORIZ 30MM).

A remarkable result is found for the convective heat transfer coefficients. Therefore, the global air circulation determined by the presence of the heat sinks with horizontal fins is also improving the heat transfer coefficients $h_c$ on the front of the PV. In this case, the $h_c$ in front of the PV panel have closer values to the ones recorded on the back. This phenomenon is less intense for the vertical fins. The wind direction seems to change the best configuration (Case 2—horizontal 30 mm) only for particular cases. The temperature reduction determined for this case is presented in Table 5.

A general tendency of the data obtained showed that the horizontal (transversal fins) are superior to those vertical (longitudinal) for each wind direction and velocity, while the best wind direction for improving the cooling of the PV panel–heat sink ensemble is towards the back of the PV panel. Also, if analyzed separately, the smaller holes ($\Phi$ = 30 mm) determine better results for horizontal fins, while the larger ones ($\Phi$ = 60 mm) are suitable for the vertical ones.

The cooling effect over the global energy production is quantified as efficiency improvement, taking into account the data available from the producer data sheet—a linear reduction of the efficiency by $-0.37\%/°C$ [32]. Therefore, for these disadvantageous external conditions ($t_{air}$ = 35 °C, $G$ = 1000 W/m$^2$), the normalized power generation was calculated—as the ratio of the maximum theoretical power $P_{STC}$ obtained in Standard Test Conditions.

The results regarding the power production are normalized and presented as a fraction of maximum theoretic power (320 Wp in STC conditions). Therefore, each degree of temperature above the standard test conditions temperature $t_{STC}$ = 25 °C determines lower efficiencies compared to the STC one ($\eta_{STC}$ = 19.30%) [32].

The visual effect over the improvement of the PV panel power generation determined by the heat sinks is presented in Figures 11–13. These figures present the normalized power productions as a ratio of the power produced in STC (considering the STC power as equal to 1). Each figure shows the amount of power produced in each case (from Case 0 to Case 6) for different wind directions (Figure 11—back wind, Figure 12—front wind and Figure 13—side wind. These charts present the results obtained for each case for the air velocity of 1 m/s. This trend is maintained for higher velocities, Table 6, but the effect compared to the base case is attenuated.

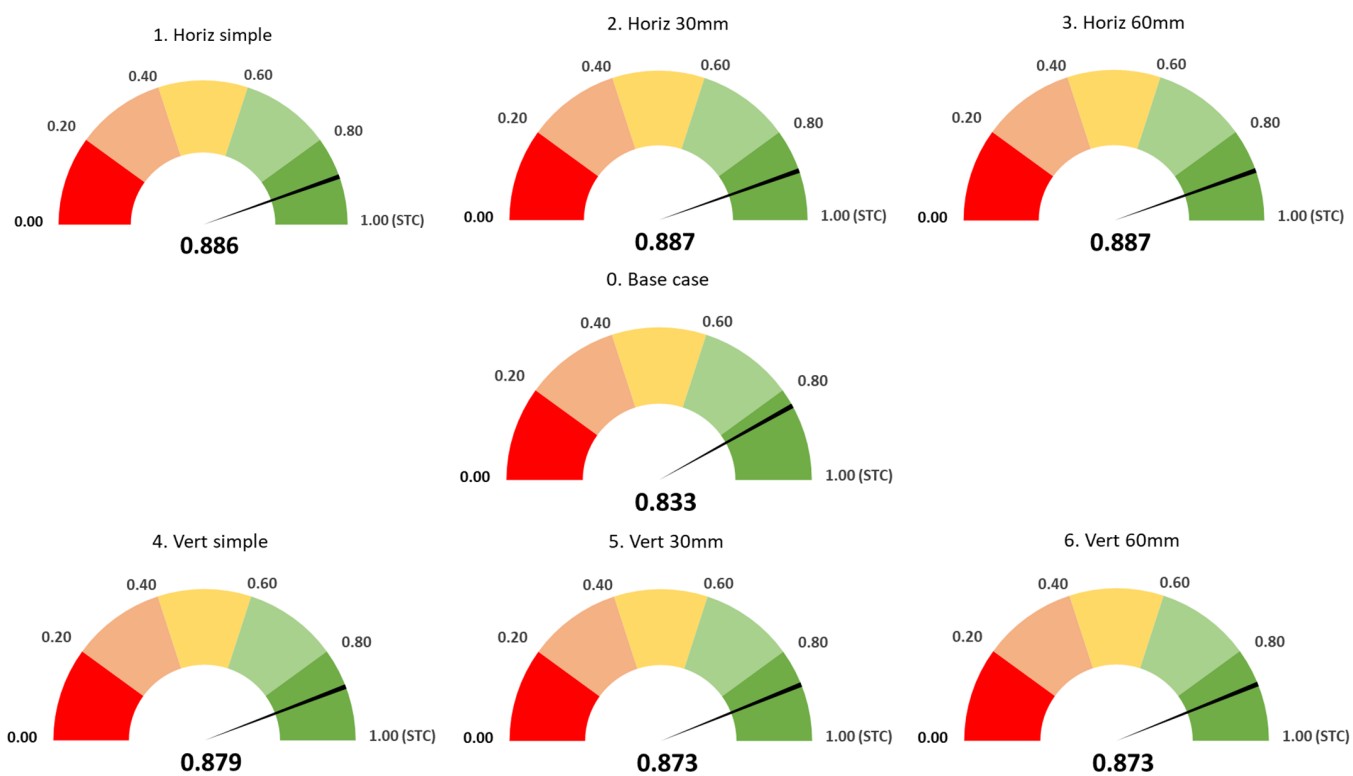

**Figure 11.** Normalized power production for **back wind**, $v_{back} = 1$ m/s, $t_{air} = 35$, $G = 1000$ W/m$^2$.

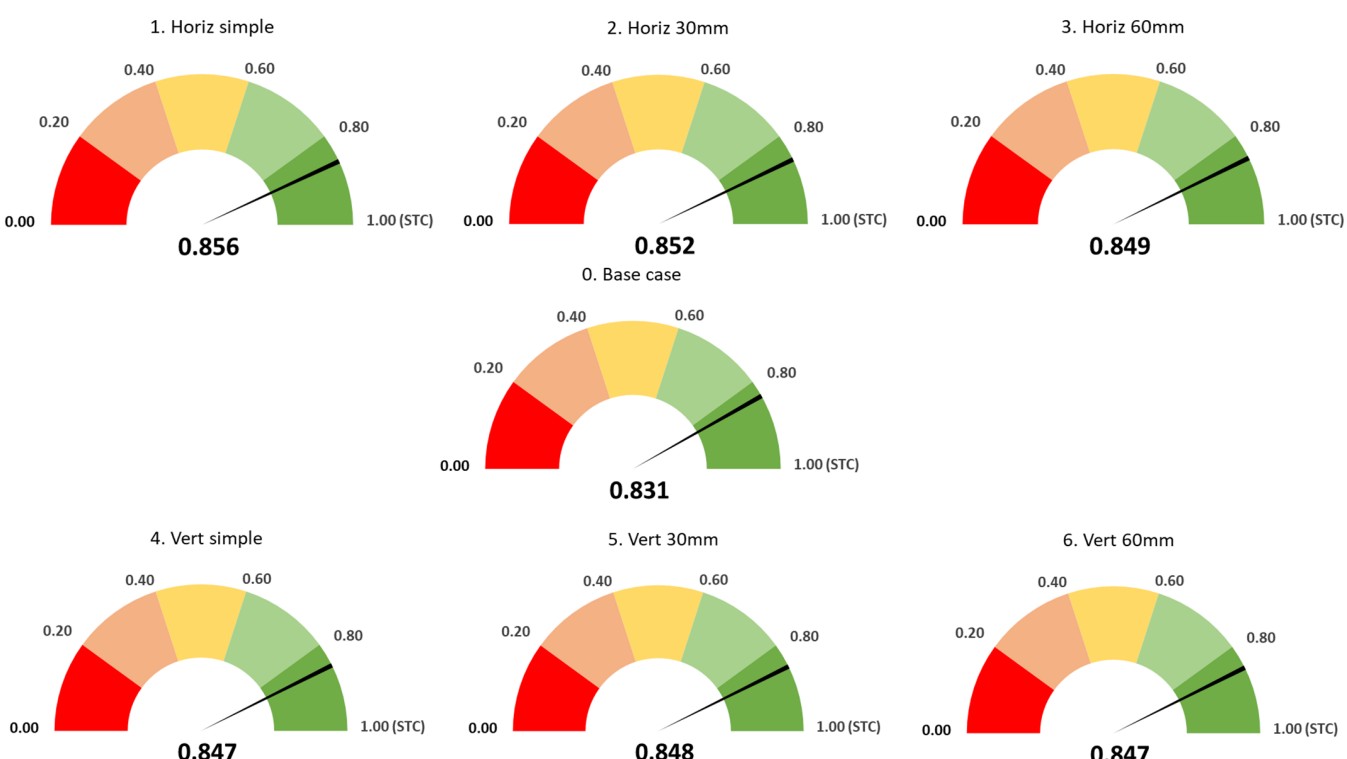

**Figure 12.** Normalized power production for **front wind**, $v_{front} = 1$ m/s, $t_{air} = 35$, $G = 1000$ W/m$^2$.

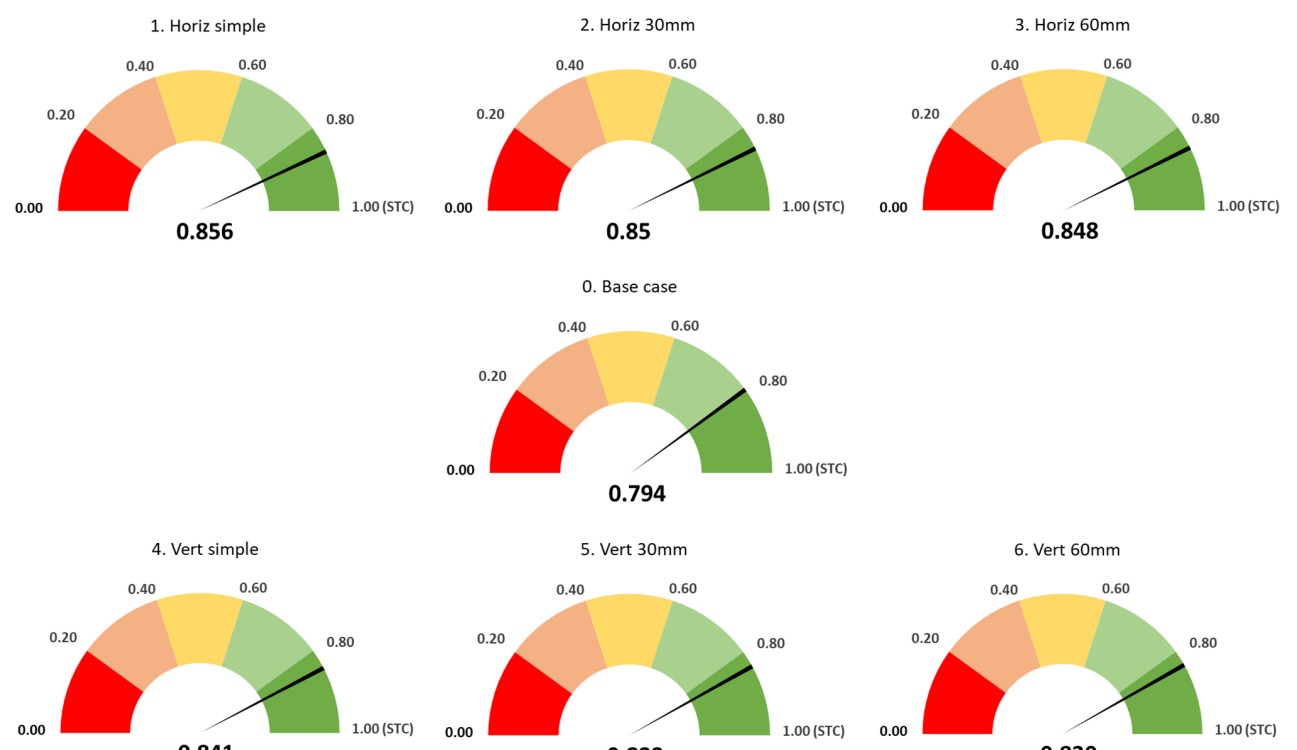

**Figure 13.** Normalized power production for **side wind**, $v_{side}$ = 1 m/s, $t_{air}$ = 35, $G$ = 1000 W/m$^2$.

**Table 6.** Normalized power production of the PV panel for each model analyzed.

| Case | $v_{air}$ | Back Wind | Front Wind | Side Wind |
|---|---|---|---|---|
| | | Normalized Power, for $t_{air}$ = 35 °C | | |
| | | $P_{STC}$ = 1.00 (320 Wp) | | |
| 0. Base Case | | 0.8333 | 0.8310 | 0.7949 |
| 1. Horiz Simple | | 0.8868 | 0.8564 | 0.8562 |
| 2. Horiz 30 mm | | 0.8874 | 0.8526 | 0.8500 |
| 3. Horiz 60 mm | 1 m/s | 0.8871 | 0.8496 | 0.8485 |
| 4. Vert Simple | | 0.8792 | 0.8473 | 0.8413 |
| 5. Vert 30 mm | | 0.8737 | 0.8489 | 0.8330 |
| 6. Vert 60 mm | | 0.8735 | 0.8478 | 0.8309 |
| 0. Base Case | | 0.8709 | 0.8697 | 0.8453 |
| 1. Horiz Simple | | 0.9111 | 0.8961 | 0.8929 |
| 2. Horiz 30 mm | | 0.9142 | 0.8982 | 0.8981 |
| 3. Horiz 60 mm | 2 m/s | 0.9104 | 0.8943 | 0.8917 |
| 4. Vert Simple | | 0.9093 | 0.8925 | 0.8865 |
| 5. Vert 30 mm | | 0.9094 | 0.8934 | 0.8835 |
| 6. Vert 60 mm | | 0.9083 | 0.8929 | 0.8848 |

**Table 6.** *Cont.*

| Case | $v_{air}$ | Back Wind | Front Wind | Side Wind |
|------|-----------|-----------|------------|-----------|
|  |  | Normalized Power, for $t_{air}$ = 35 °C | | |
|  |  | $P_{STC}$ = 1.00 (320 Wp) | | |
| 0. Base Case |  | 0.8901 | 0.8894 | 0.8636 |
| 1. Horiz Simple |  | 0.9246 | 0.9133 | 0.9097 |
| 2. Horiz 30 mm |  | 0.9260 | 0.9148 | 0.9139 |
| 3. Horiz 60 mm | 3 m/s | 0.9241 | 0.9121 | 0.9086 |
| 4. Vert Simple |  | 0.9223 | 0.9103 | 0.9053 |
| 5. Vert 30 mm |  | 0.9236 | 0.9114 | 0.9043 |
| 6. Vert 60 mm |  | 0.9205 | 0.9107 | 0.9067 |
| 0. Base Case |  | 0.9035 | 0.9028 | 0.8809 |
| 1. Horiz Simple |  | 0.9315 | 0.9226 | 0.9195 |
| 2. Horiz 30 mm |  | 0.9329 | 0.9294 | 0.9224 |
| 3. Horiz 60 mm | 4 m/s | 0.9315 | 0.9234 | 0.9181 |
| 4. Vert Simple |  | 0.9286 | 0.9228 | 0.9157 |
| 5. Vert 30 mm |  | 0.9314 | 0.9236 | 0.9160 |
| 6. Vert 60 mm |  | 0.9266 | 0.9249 | 0.9199 |
| 0. Base Case |  | 0.9132 | 0.9125 | 0.8944 |
| 1. Horiz Simple |  | 0.9359 | 0.9282 | 0.9254 |
| 2. Horiz 30 mm |  | 0.9369 | 0.9294 | 0.9282 |
| 3. Horiz 60 mm | 5 m/s | 0.9361 | 0.9275 | 0.9277 |
| 4. Vert Simple |  | 0.9334 | 0.9244 | 0.9225 |
| 5. Vert 30 mm |  | 0.9363 | 0.9294 | 0.9236 |
| 6. Vert 60 mm |  | 0.9302 | 0.9244 | 0.9232 |

Even if the highest energy production is obtained in the case of the wind direction behind the panel, for the horizontal heat sink with perforated fins, it should be specified that the most important impact in terms of cooling compared to the base case is recorded when the wind blows from the side.

It can be remarked that, for the simulated external conditions, $t_{air}$ = 35 °C, $G$ = 1000 W/m$^2$ and low air velocity $v_{air}$ = 1 m/s, the normalized power production for the base case is between 0.79 and 0.83. Meanwhile, in the best cooling conditions for each wind directions, it could reach values from 0.85 to 0.88 (Case 3—horizontal 30 mm). The improvement in power generation is between 3.06% and 7.71% compared to the base case (when no cooling is applied).

It should be highlighted that the best configuration of heat sink is the low velocity conditions for various conditions of wind velocity and direction. Therefore, Case 3—Horizontal 30 mm determines the most important effect of cooling of the PV panel. The improvement of power production is important both for low velocities (1–2 m/s) and higher ones (3–5 m/s), but the effect is lower for higher velocities (1.85–5.82%), due to the suitable cooling of the PV panel in the base case.

In terms of wind direction, the lowest cooling effect is registered for the front wind. For these cases, the presence of the heat sink does not have a significant effect, determining a raise of power production between 1.85% and 3.06% compared to the base case.

## 6. Conclusions

The goal of this study was to deploy a numerical analysis regarding the air cooling of photovoltaic panels by using heat sinks with perforated and non-perforated fins. A random PV panel with typical characteristics was analyzed for different wind directions—towards its back, front and from the side. The PV panel is mounted in fixed position, facing south, inclined at 45 degrees from the horizontal. The input constants were the solar radiation $G = 1000$ W/m$^2$ and ambient temperature $t_{air} = 35$ °C, while the parameters varied during the study were the wind direction (towards its back, front and from the side of PV panel) and velocity $v_{air} = 1$–5 m/s.

The base case of the numerical model was calibrated according to the NOCT conditions, while the accuracy of the numerical simulation was achieved by comparison with the experimental studies from literature. The numerical study is focused on six types of heat sink with perforated and non-perforated fins attached to a typical PV panel. The fins were distributed both horizontally and vertically. A challenging task consisted in simulation of the real wind conditions around the PV panel by taking into account the entire air domain. The main results obtained were the average temperature of PV panels and the respective efficiency.

The passive cooling solutions analyzed introduced a rise of maximum power production between 1.85% (wind towards the front—worse case) and 7.71% (wind towards the side—best case) above the basic set-up, depending on the wind direction and velocity. The maximum power increase for each wind direction was of 7.71% for the side wind, 6.49% for the backside wind and 3.06% for the front wind. The effect is more intense for lower velocities. This phenomenon is achieved because of the lower convection near the PV panel in the base case, compared to usage of the heat.

**Author Contributions:** Conceptualization, S.V.H., F.E.Ț. and N.-C.C.; methodology, S.V.H., F.E.Ț. and N.-C.C.; software, S.V.H. and F.E.Ț.; validation, S.V.H., F.E.Ț., N.-C.C. and I.H.; formal analysis, S.V.H., D.-A.A. and I.H.; investigation, S.V.H., F.E.Ț., N.-C.C., C.-G.P., M.V., D.-A.A. and I.H.; resources, S.V.H., F.E.Ț. and C.-G.P.; data curation, S.V.H., F.E.Ț. and I.H.; writing—original draft preparation, S.V.H., F.E.Ț., N.-C.C., C.-G.P., M.V., D.-A.A. and I.H.; writing—review and editing, S.V.H., F.E.Ț., N.-C.C., C.-G.P., M.V. and I.H.; visualization, S.V.H., D.-A.A. and I.H.; supervision, S.V.H. and M.V.; project administration, S.V.H.; funding acquisition, S.V.H. All authors have read and agreed to the published version of the manuscript.

**Funding:** This work was supported by a grant of the Romanian Ministry of Education and Research, CCCDI-UEFISCDI, project number PN-III-P2-2.1-PED-2019-1294, within PNCDI III.

**Institutional Review Board Statement:** Not applicable.

**Informed Consent Statement:** Not applicable.

**Data Availability Statement:** Not applicable.

**Acknowledgments:** We would like to acknowledge the invaluable support provided by the FCI-TUIASI infrastructure in writing this article.

**Conflicts of Interest:** The authors declare no conflict of interest.

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
