# Peer review of "Effect of Wind Direction and Velocity on PV Panels Cooling with Perforated Heat Sinks"

_applsci, doi:10.3390/app12199665_

Round 1

Reviewer 1 Report

Thank you for submitting your paper “Effect of Wind Direction and Velocity on PV Panels Cooling 2 with Perforated Heat Sinks” to the Journal of Applied Sciences.

 The goal of this study was to deploy a numerical analysis regarding the air cooling of photovoltaic panels by using heat sinks with perforated and non-perforated fins. A random PV panel with typical characteristics was analysed for different wind directions towards its back, front and from the side.

 The article is well written and well structured as a scientific text; the literature review is comprehensive and up to date. However, several issues need to be addressed properly before the paper is considered for publication.

 3.2. Numerical Modeling

 The work definitely needs to be strengthened on the following points. Validation of the calculation grid is needed with a sensitivity analysis of grid spacing. A validation of the cell size and any time step used is needed, using " grid convergence index". You can found an example in

 https://doi.org/10.1016/j.energy.2016.06.002

https://doi.org/10.3390/en7128465

 Regarding Figure 3, it is necessary to integrate it by indicating the boundary conditions for each face of the calculation volume.

 Has the air gas been set to constant density?

 The authors neglected the relative humidity of the air, can they make a rough estimate of the percentage by which it might affect the final results?

 It might be interesting to extend the analysis to lower tilt values as low as 30°C, which are more typical of Mediterranean areas.

Author Response

Dear reviewer,

Thank you for your kind remarks regarding our work! Please find attached our responses to your observations.

Reviewer 2 Report

1. Validation of the numerical model using NOCT conditions and Equation 1 is appropriate. Validation of the numerical simulation - base case—PV panel without heat sink is appreciated. Validation of the numerical simulation—PV panel with heat sink is appreciated. 

2. Numerical simulations are validated using experimental work in the literature, appreciated.

3. Results and discussions have to be improved. For example in figures 8-10, it is not clear about 35, 45, 55,65,75 and 85. Units are not given and they are not discussed in the manuscript.

4. Operating temperature of PV panel for studied cases for back wind, side wind anf front wind are good and compared well

5. Figures 11-12 are depicted well but the details on the figures are missing. It is not clear about those values mentioned in the figures

6. Normalized power production of the PV panel for each model analyzed, done well

7. Authors have concluded well. The abstract and conclusions are well written, appreciated

What has to be addressed:

as mentioned above,

a) Results and discussions have to be improved. For example in figures 8-10, it is not clear about 35, 45, 55,65,75 and 85. Units are not given and they are not discussed in the manuscript.

b) Figures 11-12 are depicted well but the details on the figures are missing. It is not clear about those values mentioned in the figures.

Author Response

(The authors gave the same response as above.)

Round 2

Reviewer 1 Report

Authors addressed the reviewers' suggestions